# Chemometric Approaches to Analyse the Composition of a Ewe’s Colostrum

**DOI:** 10.3390/ani13060983

**Published:** 2023-03-08

**Authors:** Massimo Todaro, Giuseppe Maniaci, Riccardo Gannuscio, Daniela Pampinella, Maria Luisa Scatassa

**Affiliations:** 1Dipartimento SAAF, Università di Palermo, 90128 Palermo, Italy; 2Istituto Zooprofilattico Sperimentale della Sicilia, 90129 Palermo, Italy

**Keywords:** colostrum composition, dairy ewes, correlations

## Abstract

**Simple Summary:**

Colostrum is the first food consumed by infants, and it provides a source of passive immunity for newborns due to its concentration of immunoglobulins, which do not enter an embryo’s bloodstream in cattle, sheep, goats, and horses. In this study, 60 bulk colostrum samples were collected from Valle del Belice ewes. The samples were analysed for their gross composition, pH, Brix refractive index, colour, fatty acid composition, and mineral content. The objective of this study was to evaluate the correlation patterns between the bulk tank colostrum parameters and to define, using a multivariate approach, the relationships between them. This study confirmed that the Brix refractive index is a good parameter for simply evaluating the nutritional quality of sheep colostrum on a farm. Understanding the correlations between the compositional parameters will allow for interventions to correctly feed lambs in their first days of life.

**Abstract:**

Colostrum is a major source of immunity in lambs and, in general, in all newborn ruminants. It allows the transfer of antibodies from the ewe to the lamb, and it becomes the exclusive source of nutrients for a newborn. Among the most significant Pearson correlations, the positive correlation between the Brix refractive index (Brix) and protein (0.90) should be noted. Both parameters (protein percentage and Brix) were then positively correlated with the percentage of fat (0.38 and 0.41), urea (0.81 and 0.67), calcium (0.39 and 0.29), and magnesium (0.58 and 0.59), as well as the yellowness (0.78 and 0.75). Somatic cell count (SCC) and pH, parameters which are indicators of subclinical mastitis, were positively correlated (0.49), and SCC was positively correlated with sodium (0.37) and negatively correlated with potassium (−0.28). Among the macroelements in colostrum—calcium, potassium, magnesium, and sodium—the correlations were largely positive. With respect to the colour parameters, yellowness was negatively correlated with lightness (−0.41) and redness (−0.45). The factor analysis split the total of variance into three latent factors. The first factor was named “Colostrum quality of grazing sheep” because it was positively correlated primarily with SCC, pH, Poly Unsatured Fatty Acids (PUFA), and the sum of the omega-3 Fatty Acids (FAs). The second factor, named “Good quality colostrum”, was positively correlated primarily with the refractive index, protein and fat percentages, urea content, phosphorus, Mono Unsatured Fatty Acids (MUFA), and yellowness. The third factor was positively correlated primarily with calcium, potassium, magnesium, and sodium, and it was therefore termed “Mineral component of colostrum”. Stepwise discriminant analysis showed that the protein percentage, calcium, and magnesium were able to explain more than 85% of the Brix refractive index, which remains a good parameter for simply evaluating the nutritional quality of sheep colostrum at the level of a farm.

## 1. Introduction

Colostrum is nature’s plentiful surplus source of nutrient-bioactive rich, viscous fluid secreted after parturition during the first 72 h, and it is highly dynamic in a modest volume format. Biologically important bioactive proteins, immunoglobulins (e.g., IgG, IgM, and IgE), lactoferrin, growth factors, hormones, cytokines, interleukins, nucleosides, and nucleotides are the main bioactive components of colostrum. These constituents are also much more prominent in the first hours after lambing. Moreover, colostrum is a sufficient nutritional source of proteins, fatty acids (FAs), vitamins, and minerals, all of which are essential for a wide range of metabolic processes [1]. The composition and quality of colostrum are influenced by many factors prior to parturition, including changes in diet availability and composition [2,3,4], hormone production [5], the timing of colostrum let-down [6], colostrum quality [7,8,9], and viscosity [10,11]. These factors subsequently impact lamb survival rates, where delayed let-down, highly viscous colostrum or abnormal configuration, and damage to the udders and teats reduce the colostrum intake of a lamb [12,13].

Colostrum has been a great area of interest in pigs and cattle for many years, and in comparison, research on sheep colostrum is limited, especially with regard to measuring and defining colostrum quality [13].

Colostrum quality is primarily measured by laboratory assays tests in which the IgG content is determined using an ELISA test or, more commonly, a radial immunodiffusion assay (RID) [14]. These laboratory methods, although considered the best for the determination of IgG immunoglobulins, require specialized equipment and high execution times for determination [15]. One tool for easy determination of colostrum quality is a portable digital refractometer, the Brix refractometer, which allows for a quick, real-time estimate of the percentage of total solids in colostrum (Brix). Although the Brix refractometer has been widely used and validated for dairy cattle [16], it has only recently been used to assess the quality of sheep colostrum [17,18,19,20], where the “quality” is also defined by the concentration of IgG inside the colostrum. The quality of the colostrum is also influenced by the constituents of the composition such as fats, lactose and proteins; however, few studies have shown the relationship between the Brix refractive index and these constituents [17,18,20].

In a previous paper [16], we reported the effect of lambing season on Valle del Belice ewes’ colostrum composition, while the objective of our current study was to evaluate the correlation patterns between Brix refractive index, bulk tank colostrum parameters (gross composition, fatty acids, minerals, and colour) and to define, using a multivariate approach, the relationships between them.

## 2. Materials and Methods

Colostrum samples were collected as part of routine animal colostrum collection in breeding farms, and is a non-experimental, veterinary practice. No animal discomfort was caused for sample collection for the purpose of this study. The Directive 2010/63/EU of the European parliament and the council and the Italian D.lgs 26/2014 do not apply to non-experimental practices. An ethical review by the Animal welfare body was therefore not required.

### 2.1. Sampling

A total of 60 pooled samples of colostrum were collected in two seasons of lambing: the summer season of lambing (SSL), with sampling occurring from the end of August to the beginning of October, and the winter season of lambing (WSL), with sampling occurring from the end of December to the beginning of February. Every morning, for pluriparous lambings (i.e., ≥2 lambs born to a ewe), the colostrum was hand-milked completely, filtered, and mixed in the same quantity and in the same container. After this, 50 mL of colostrum was placed in each of 6 sterile bottles and immediately frozen at −20 °C. The residual colostrum was immediately given to the lambs with a baby bottle. For each season of lambing (SSL and WSL), 30 pooled samples of colostrum were collected.

The SSL and WSL were very different in terms of the feeding and management of pregnant ewes. During the SSL, pregnant ewes were confined in fences near the sheepfold and fed with hay ad libitum, in addition to a concentrate (300 g/head/d of barley grain). In the WSL, the pregnant ewes grazed with the non-lactating ewes primarily on the sulla meadows (*Sulla coronaria* L.) due to the good availability of pasture, and no hay or concentrates were used.

### 2.2. Physicochemical Analyses

After all colostrum samples were thawed to room temperature (20–25 °C), the refractive index was measured in duplicate with an optical Brix refractometer (Manual Refractometer MHRB-40 ATC, Mueller Optronic, Erfurt, Germany). The refractometer was equipped with a Brix scale that ranged from 0 to 40% Brix. The accuracy of the instrument was ±0.2% Brix at 20 °C. To detect the chemical parameters, the colostrum samples were diluted with deionized water (1:1) and vortexed for 10 s to ensure adequate homogeneities, and then the samples were warmed at 42 °C in a water bath and analysed for lactose, fat, protein, casein, urea via the infrared method, and somatic cell count (SCC) via flow cytometry (Combi-Foss 6000, Foss Electric, Hillerød, Denmark). The SCCs were logarithmically transformed to normalize their distribution. The obtained results were recalculated considering the dilution factor. In the undiluted samples, the pH and colour parameters were determined. The pH levels were measured using an HI 9025 pH meter (Hanna Instruments, Ann Arbor, MI, USA). The colour parameters were measured using a Minolta Chroma Meter CR-300 with illuminant C (Minolta, Osaka, Japan) according to CIE LAB method [21] and included lightness (L*, from 0 = black to 100 = white), redness (a*, from green = −a to red = +a), and yellowness (b*, from blue = −b to yellow = +b). The colour parameters were measured by placing the lens directly over a capsule containing the colostrum sample. Colour was assessed in triplicate, and so the results reported are the averages of three measurements of the same sample.

### 2.3. Fatty Acids

The lyophilized colostrum samples were utilized for fatty acid analysis. A total of 100 mg of freeze-dried colostrum sample was directly methylated in 1 mL hexane with 2 mL 0.5 M NaOCH_3_ at 50 °C for 15 min, followed by 1 mL 5% HCl in methanol at 50 °C for 15 min based on the bimethylation procedure [22]. Fatty acid methyl esters (FAMEs) were recovered in 1.5 mL hexane. Using an auto sampler, 1 μL of each sample was injected into an HP 6890 gas chromatography system equipped with a flame-ionization detector (Agilent Technologies, Santa Clara, CA, USA). The FAMEs were separated using a CP-Sil 88 capillary column (100 m long, 0.25 mm internal diameter, and a 0.25 µm film thickness; Chrompack, 163 Middelburg, The Netherlands). The injector temperature was kept at 25.5 °C, and the detector temperature was kept at 250 °C, with a hydrogen flow of 40 mL/min, an air flow of 400 mL/min, and a constant helium flow of 45 mL/min. The initial oven temperature was held at 70 °C for 1 min and then increased by 5 °C/min until it reached 100 °C, where it was held for 2 min, and then it was increased by 10 °C/min to 175 °C, held for 40 min, and, finally, increased by 5 °C/min to a final temperature of 225 °C and held for 45 min. Helium, with a pressure of 158.6 kPa and a flow rate of 0.7 mL/min (linear velocity 14 cm/s), was used as a carrier gas. A FAME hexane mix solution (Nu-Check-Prep, Elysian, MN, USA) was used to identify each FA. To quantify the total FA, one millilitre of C23:0 solution (20 mg/50 mL of hexane, Sigma-Aldrich, St. Louis, Missouri, USA) was used as the internal standard. In a previous paper, the complete colostrum FA composition was reported [16], while in this study, only monounsaturated FAs (MUFA), polyunsaturated FAs (PUFA), saturated FAs (SFA), and total ω-3 and ω-6 FAs were reported.

### 2.4. Minerals

Using a microwave plasma atomic emission spectrometer (Agilent 4200 MP AES; Agilent Technologies, Santa Clara, CA, USA), the mineral composition of the colostrum samples was determined after acid digestion had been carried out using a CEM MARS XPRESS 230/60 Microwave Accelerated Reaction System, CEM Corporation 3100 Smith Farm Road Matthews, NC 28104.

Briefly, 1 mL of each fresh colostrum sample was transferred to a 15 mL Teflon digestion vial. Before capping the vials, 3 mL of nitric acid and 1 mL of hydrogen peroxide were added to each sample. We also prepared a blank solution containing 3 mL of nitric acid and 1 mL of hydrogen peroxide. Analytical-grade concentrated nitric acid (HNO_3_ 67–69%) and hydrogen peroxide (H_2_O_2_ 30%) were used for the samples’ digestion. Each colostrum sample and the blank solution were detected in triplicate. The microwave digestion of the samples was carried out at a power of 1200 W and at a temperature of 200 °C, which was reached in 10 min. These values were maintained for 30 min (i.e., the duration of the digestion). Upon completion of the program, each digested sample was diluted to a final volume of 12 mL with 18.2 MΩ deionized water and then diluted an additional 10 times with a solution of 2% nitric acid. In a previous paper, the macro- and micro-elements were reported [16], while in this study, only the macro-elements (Ca, P, K, Mg, and Na) were considered.

### 2.5. Statistical Analysis

Three different approaches were used for the statistical analysis of the detected data: simple correlation analysis (CORR), stepwise discriminant analysis (SDA), and multivariate factor analysis (MFA). CORR is a statistical method that is used to discover if there is a relationship between two variables and to measure how strong that relationship may be. The SDA was used to evaluate the variables that better explained the colostrum Brix refractive index. Forward selection was used with the most significant predictor at each step being added to the model. The significance threshold for entry into the model was *p* = 0.15. The data relative to the 19 colostrum parameters (all except SFA, which made the matrix singular) were analysed via MFA. The number of factors to be extracted was based on their eigenvalue (>1) [23], on their readability in terms of their relationships with the original variables and their biological meanings, and on the amount of explained variance. Factor interpretation was improved through a VARIMAX rotation. VARIMAX is an orthogonal rotation which is based on the maximization of the sum of the squares of the factor loadings [24,25]. SAS software (version 9.1, SAS Institute Inc., Cary, NC, USA), proc CORR, REG, and FACTOR were utilized.

## 3. Results

### 3.1. Descriptive Statistics

Simple statistics of the colostrum quality are reported in Table 1. The mean of the Brix refractive index was 21.02 ± 1.98%. The means of the gross compositions of the Valle del Belice ewes’ colostrum were 12.10 ± 1.73%, 8.01 ± 1.51%, and 3.13 ± 0.45% for protein, fat, and lactose percentages, respectively. The somatic cell counts presented a logarithm value of 5.81 that corresponded to 645,000 cells per mL of colostrum, and the mean pH was 6.36 ± 0.18.

The most abundant minerals detected in the ewes’ colostrum were phosphorus (82.87 ± 10.99), calcium (26.26 ± 6.28), potassium (15.11 ± 3.39), sodium (13.42 ± 5.52), and magnesium (5.06 ± 1.47). The fatty acid compositions of the colostrum shown as saturated FAs were the most abundant (SFA, 59.43 ± 3.24) with respect to monounsaturated fatty acids (MUFA, 31.77 ± 2.50) and polyunsaturated fatty acids (PUFA, 8.80 ± 2.66) FAs, the percentages of omega-3 (1.11 ± 0.53) and omega-6 (3.01 ± 0.45) FAs were interesting. The CIE LAB parameters of the colostrum presented as a reddish-yellow colour (L* = 84.09; a* = −7.87; b* = 19.03).

### 3.2. Pearson Correlation Analysis

The Pearson correlations between the 20 colostrum variables considered in this study are reported in Table 2 and Table 3.

With respect to the gross composition, the most important parameter of the colostrum quality, the protein percentage, was positively correlated with urea (0.81) and fat (0.38) and negatively correlated with lactose (−0.41). Moreover, positive correlations were found with calcium (0.39) and magnesium (0.58), while the relationships with FAs showed a positive correlation with MUFA (0.34) and a negative correlation with SFA (−0.32). Finally, the colostrum protein percentage was positively correlated with yellowness (0.78) and negatively correlated with lightness (−0.45) and redness (−0.37). The highest and most significant correlation was with protein percentage, which was detected with a Brix refractive index of 0.90; therefore, all the correlations between the Brix refractive index and the other parameters were similar to those detected and reported above for the protein percentages.

Fat percentage was positively correlated with redness (0.32) and yellowness (0.42). Lactose percentage was negatively correlated with SCC (−0.53), pH (−0.39), sodium (−0.55), and yellowness (−0.54), and it was positively correlated with potassium (0.30). SCC and pH were positively correlated (0.49), and analogous correlations were detected with sodium (0.37), while there was a negative correlation between SCC and potassium (−0.28). Between minerals, the correlations were largely positive, in particular, between calcium and potassium (0.64) and magnesium (0.78) and sodium (0.31). With respect to the correlations between PUFA, MUFA, and SFA, the values detected between them were obvious, and the positive correlations found between PUFA and pH (0.55) and PUFA and SCC (0.48) were interesting. With respect to the colour parameters, yellowness was negatively correlated with lightness (−0.41) and with redness (−0.45). Furthermore, the latter was found to be positively correlated with SCC (0.44) and pH (0.44).

### 3.3. Multivariate Analysis

The SDA selected the following most-discriminant eight variables: protein percentage, calcium, magnesium and sodium content, lactose percentage, yellowness, the sum of the omega-6 FAs, and the potassium content (Table 4).

Protein percentage was the variable that contributed the most to the determination coefficient (R^2^) of the regression model (0.805), and the other seven variables entered into the model (*p* < 0.15) explained only 10% of the variability. Altogether, the eight colostrum components explained 90.8% of the variability of the colostrum refractive index.

The results of the MFA are reported in Table 5. Three factors were retained by the factor analysis, according to the amount of variance explained and to their interpretation in terms of the biological meaning and their relationships with the original variables. Their structure, after the VARIMAX rotation, was easy to read. Each factor exhibited few large and many small loadings, and each variable had a large loading in only one factor and small loadings in the others, respectively.

The first factor (which explained 68.4% of the original variance) was correlated primarily with SCC, pH, PUFA, and the sum of the omega-3 FAs, and it was negatively associated with the sum of the omega-6 FAs. The ewes’ colostrum samples with higher scores for this factor were richer in SCCs and presented higher pH values. In addition, the PUFA and the sum of the omega-3 FAs were positively associated with this factor; therefore, could be termed “Colostrum quality of grazing sheep”. The second factor (which explained 23.3% of the original variance) was positively correlated primarily with the refractive index, protein and fat percentages, urea content, phosphorus, MUFA, and yellowness, and it was negatively associated with lactose percentage, lightness, and redness. The colostrum samples of the ewes with higher scores of the second factor were richer in protein content, presented higher refractive index values, and appeared to be more yellow, which are elements that indicate colostrum with better nutritional quality. Therefore, this factor could be named “Good quality colostrum”. The third factor (which explained 8.3% of the original variance) was positively correlated primarily with calcium, potassium, magnesium, and sodium; therefore, the colostrum samples that had higher scores for the third factor were richer in macro-elements; therefore, could be termed “Mineral component of colostrum”.

## 4. Discussion

### 4.1. Simple Statistics

The Brix refractometer enables the rapid on-farm estimation of colostrum quality. This method has been intensively studied in bovines and was validated for sheep colostrum by Kessler et al. [17], demonstrating a correlation coefficient between the IgG and Brix values equal to 0.75, while Santiago et al. [20] reported that the Brix refractometry provided a satisfactory estimate of the total protein present in Santa Inês ewes’ colostrum samples, as well as in the blood serum samples of lambs at different times after lambing. The mean Brix value detected in the Valle del Belice ewes’ colostrum samples was comparable with those reported by other papers [17,20]. The gross composition of the Valle del Belice ewes’ colostrum showed high protein and in fat percentages with low lactose content, and analogous compositions have been reported by other authors for Italian [26,27] and foreign sheep breeds [17,28]. The mean values of the colostrum SCC and pH were similar to those reported by other authors [27,29,30,31], and in particular, the pH values detected in the colostrum samples were lower than those of milk and increased with time, post-partum [31]. The precise reason for the low pH of colostrum is unknown, and during pregnancy, there is increased permeability of the mammary gland membranes, and thus, more blood constituents gain access to the milk; therefore, a pH closer to that of blood (pH of 7.35 to 7.45) would be expected. Sebela and Klicnik [32] reported that the low pH of colostrum is caused by the increased concentration of protein, dihydrogen phosphate, citrate, and carbon dioxide. In a previous study [16], it was reported that there was a significant influence of the season of lambing on SCC and pH, which presented higher values in the colostrum samples when collected during the WSL with respect to the SSL. The authors explained that this fact was due to the WSL, in which the pregnant ewes grazed in the pasture until lambing, leading to greater rubbing of the udders and thereby increasing the SCC value due to the higher levels of flaking epithelial cells. 

In the literature, only a few papers have reported on the minerals in sheep colostrum [26,33], and the macro-elements detected in the colostrum samples were comparable to the data reported by Kráčmar et al. [33] for ewes reared on an organic farm in the Czech Republic.

The colostrum FA composition confirmed that SFAs are the most abundant FAs, and similar percentages (58%) were found by Guiso et al. [26] for Sarda ewes’ colostrum samples and (61%) by Martini et al. [27] for Massese ewes’ colostrum samples, while a lower percentage (50%) was reported for Saudi Arabian ewes [34], and a higher percentage of SFAs (69%) was reported for Rideau-Arcott ewes reared in Canada [35]. The highest percentage of PUFAs detected in the Valle del Belice ewes’ colostrum samples, as compared to other breeds [26,27,34,36], may be due to their genetic diversity, but in our opinion, the most important factor was the feeding of the pregnant ewes. In fact, Todaro et al. [16] reported that the lambing season significantly influenced the FA profiles of ewes’ colostrum, indicating that PUFA percentages increase during the winter season, when green forage represents the most prevalent, if not exclusive, component in the diets of pregnant ewes. The omega-6/omega-3 ratio was favourable for the good presence of omega-3 FAs, and similar results have been reported for other sheep breeds’ colostrum samples [26,36].

The CIE LAB parameters detected in our colostrum samples showed a reddish-yellow colour, which may have been due to the presence of retinol and xantophylls [31], while the carotenoids resulting from the ewes’ diets (fresh forage) were converted into vitamin A, which lacks colour [35], and so the yellow colour was likely due to the presence of a large amount of fat [31].

### 4.2. Pearson Correlation Analysis

The simple correlation between protein and other colostrum parameters showed a strong correlation with urea and fat content, in accordance with those found in the Valle del Belice ewes’ milk samples [37]. The positive correlations between colostrum protein percentage and calcium and magnesium were consistent with the results of other studies on ewe milk [38]. Colostrum protein percentage was positively correlated with MUFA and negatively with SFA, and in the literature, positive correlations were found between milk protein percentage and MUFA or SFA both for cow milk [39] and goat milk [40].

With respect to the correlation between colostrum gross composition and colour parameters, fat and protein percentages were positively correlated with yellowness while only protein was negatively correlated with lightness, and analogous results were found by other researchers for cow colostrum [41,42]; thus, colostrum with a more yellow and darker colour is very likely to contain more fat and protein. Moreover, the correlations found between the Brix values and the colour parameters showed that when the colostrum is more highly coloured, it presents a higher Brix value (a positive correlation between the Brix value and yellowness and a negative correlation with redness), with a probable greater IgG content; in addition, retinol and α-tocopherol content may be higher in colostrum that is more red and yellow in colour [42]. Between the CIE LAB parameters, significant negative correlations were only found between yellowness and lightness and redness, and an analogous correlation between L* and b* (r = −0.25) was found for cow colostrum, while an opposite sign was reported as the correlation between the a* and b* parameters (r = 0.41) [42].

The Brix value, which is considered to be an inexpensive, rapid, and satisfactorily accurate method for estimating IgG concentrations [43], was positively correlated with the percentages of protein and fat, confirming that the higher the Brix value, the better the quality of the colostrum.

The correlations between the parameters linked to udder health showed that SCC and pH varied jointly with a positive correlation, and both parameters were positively correlated with sodium and negatively with potassium. When the SCC increased, the presence of subclinical mastitis was hypothetical; therefore, in this case, the blood–milk barrier was damaged and the mammary epithelial tight junctions became leaky, leading to the escape of blood and components into the lumen of the alveoli, which produced an increase in colostrum SCC and pH values [44,45]. In this scenario, sodium and chloride (which are more present in extracellular fluid) pour into the lumen of the alveolus, and in order to maintain the osmolarity, potassium levels decrease proportionately [46]. A significant negative correlation was found between the percent of lactose and SCC content and pH, highlighting that a high SCC content is associated with a low milk yield and a low lactose content, as has frequently been reported in the literature [47]. Moreover, SCC and pH were positively correlated with PUFA and the sum of the omega-3 FAs. This finding was explained by Todaro et al. [16], who found that in WSL colostrum, when PUFAs and omega-3 FAs were more abundant, pregnant ewes grazed until lambing, causing more udder rubbing between the legs and, therefore, probable higher incidences of traumatic stress. Additionally, the positive correlation between a* and SCC or pH could be explained from trauma of the mammary gland [48].

The mineral components of the colostrum exhibited several positive correlation coefficients with each other, and most of these were consistent with a past study of cow colostrum samples collected in India [38].

### 4.3. Multivariate Analysis

The coefficient of correlation is not informative in all cases in explaining cause-and-effect relationships in the variables, because the association between two variables may depend on a third variable. The use of SDA provides a plausible explanation of observed correlations by modelling the cause-and-effect relations among the variables [49].

The stepwise discriminant analysis confirmed that there was a strong relationship between the refractive index and the colostrum protein percentages, which alone is able to explain more than 80% of the variability in the Brix value. Sodium, calcium and magnesium were the other three variables that explained a further 6% of the total variability, and with all the other variables entered into the model, this was explained further, but they did not have a very significant share of the variability. The SDA showed a strong relation between Brix value, protein percentages and the mineral component of colostrum. Based on actual knowledge, the refractive index remains an acceptable tool for on-farm estimations of colostrum quality in does and ewes, despite the distinct between-species variations in their colostrum compositions [17].

The adequateness of the factor analysis for fitting the colostrum components correlation matrix was confirmed by the simple structure of the rotated pattern [23]. In particular, each factor showed large loadings with few variables and small loadings with the other variables, respectively. Each variable had a large loading in only one factor, with only two exceptions: the magnesium content and the a* colour parameter. The first factor was named “Colostrum quality of grazing ewes” because it was positively correlated with SCC, pH, PUFA, and the sum of the omega-3 FAs, and it was negatively correlated with the sum of the omega-6 FAs. As mentioned earlier during the correlations discussion, the ewes that grazed during the WSL, with diets rich in fresh forage and Sulla, in particular, presented colostrum samples that were richer in PUFAs and omega-3 FAs [16]. Moreover, this type of breeding of pregnant ewes resulted in increased wear of the udders, which likely resulted in damage to the tight epithelial junctions and, consequently, in higher SCC and pH values in the colostrum. The second factor was called “Good colostrum quality” because it was positively correlated with all the variables that identified good quality colostrum, such as the Brix value, the percentages of proteins and fats, the content of phosphorus and magnesium, the presence of MUFAs, and a yellow colour. The third factor was named “Mineral component of colostrum” because it was found to be positively correlated with the primary macro-elements found in sheep colostrum: calcium, potassium, magnesium, and sodium. Few studies are present in the international literature about ewe colostrum composition, and little is known about the colostrum mineral concentrations in different dairy breeds. Moreover, there is growing evidence of the involvement of minerals on the immune and antioxidant responses that are key for preserving newborns’ health, and there is interest in trace mineral supplementation to maximize these positive effects in calves [50]. To understand the relationships and the correlations between the minerals and other constituents of colostrum will allow for a targeted intervention in the mineral integration of young ruminants.

## 5. Conclusions

The different statistical approaches used in this study were able to efficiently summarize the ewes’ colostrum profiles with a reduced number of new variables (latent factors). In particular, some independent factors have been associated with the metabolic pathways involved in the synthesis and modification of colostrum, defining the factors related to nutritional quality.

This study confirmed that the Brix refractive index is a good parameter for simply evaluating the nutritional quality of sheep colostrum on a farm. The correlation analysis between the physical and chemical parameters of the colostrum samples allowed to identify the mechanisms that regulate the biological production of colostrum; therefore, researchers will be able to intervene to provide the correct nutrition for the lambs.

## Figures and Tables

**Table 1 animals-13-00983-t001:** Physicochemical parameters of ewe’s Colostrum.

Parameters	Mean Value	Standard Deviation	Minimum Value	Maximum Value
Gross composition				
Brix (%)	21.02	1.98	16.80	27.00
Protein (%)	12.10	1.73	9.16	16.44
Urea (mg/dL)	47.94	14.44	21.02	81.36
Fat (%)	8.01	1.51	4.18	11.52
Lactose (%)	3.13	0.45	1.76	4.28
Somatic Cell Count (Log10)	5.81	0.72	4.60	7.33
pH	6.36	0.18	6.06	6.82
Minerals				
Calcium, C (mmol L^−1^)	26.26	6.28	11.74	45.35
Phosphorus, P (mmol L^−1^)	82.87	10.99	51.12	109.38
Potassium, K (mmol L^−1^)	15.11	3.39	6.40	22.04
Magnesium, Mg (mmol L^−1^)	5.06	1.47	2.20	10.93
Sodium, Na (mmol L^−1^)	13.42	5.52	4.18	32.02
Fatty acids (FA)				
Saturated fatty acids (g/100 g FA)	59.43	3.24	53.57	69.08
Monounsaturated fatty acids (g/100 g FA)	31.77	2.50	25.75	36.52
Polyunsaturated fatty acids (g/100 g FA)	8.80	2.66	5.17	13.68
∑ ω-3 (g/100 g FA)	1.11	0.53	0.65	2.65
∑ ω-6 (g/100 g FA)	3.01	0.45	2.12	4.02
Color				
Lightness (L*)	84.09	2.39	76.05	90.26
Redness (a*)	−7.87	1.00	−9.66	−4.58
Yellowness (b*)	19.03	2.81	13.43	24.62

**Table 2 animals-13-00983-t002:** Pearson correlation coefficients between physicochemical parameters of ewe’s Colostrum.

	BRIX	PRT	UREA	FAT	LACT	SCC	pH	Ca	P	K	Mg	Na
BRIX	1											
PRT	0.90 ***	1										
UREA	0.67 ***	0.81 ***	1									
FAT	0.41 ***	0.38 ***	0.13	1								
LACT	−0.37 ***	−0.41 ***	−0.17	−0.52 ***	1							
SCC	0.17	0.20	0.11	0.31 **	−0.53 ***	1						
pH	−0.18	−0.12	−0.14	0.21	−0.39 **	0.49 ***	1					
Ca	0.29 *	0.39 ***	0.36 **	0.05	−0.10	−0.01	−0.12	1				
P	0.27 *	0.22	0.07	0.08	−0.02	0.01	−0.14	0.07	1			
K	0.16	0.18	0.22	−0.13	0.30 **	−0.28 *	−0.48 ***	0.64 ***	−0.03	1		
Mg	0.59 ***	0.58 ***	0.44 ***	0.16	−0.20	−0.03	−0.26 *	0.78 ***	0.11	0.64 ***	1	
Na	−0.10	0.02	0.08	0.09	−0.55 ***	0.37 **	0.40 ***	0.31 **	−0.24 *	−0.01	0.32 **	1
PUFA	0.01	0.06	−0.08	0.16	−0.14	0.48 ***	0.55 ***	0.02	−0.07	−0.21	−0.21	−0.08
MUFA	0.29 *	0.34 **	0.14	0.14	−0.08	−0.15	−0.27 *	0.12	0.29 *	−0.05	0.25 *	−0.04
SFA	−0.23 *	−0.32 **	−0.04	−0.24 *	0.17	−0.27 *	−0.24 *	−0.11	−0.17	0.22	−0.02	0.10
∑ ω-3	−0.05	−0.04	−0.18	0.19	−0.05	0.48 ***	0.48 ***	0.04	−0.03	−0.17	−0.22	−0.14
∑ ω-6	0.12	0.17	0.25	−0.06	0.04	−0.30 **	−0.30 **	0.16	0.14	0.07	0.34 **	0.20
L*	−0.32 **	−0.45 ***	−0.42 ***	0.10	0.16	−0.15	−0.15	0.06	0.16	0.03	−0.15	−0.09
a*	−0.34 **	−0.37 **	−0.36 **	0.32 **	−0.07	0.44 ***	0.44 ***	−0.27 *	−0.14	−0.25 *	−0.34 **	0.18
b*	0.75 ***	0.78 ***	0.48 ***	0.42 ***	−0.54 ***	−0.03	−0.03	0.17	0.17	0.06	0.48 ***	−0.02

PRT: protein; LACT: Lactose; SCC: Somatic Cell Count (Log10); PUFA: Polyunsaturated fatty acids; MUFA: Monounsaturated fatty acids; SFA: Saturated fatty acids; L*: Lightness; a*: Redness; b*: Yellowness; * *p* < 0.05; ** *p* < 0.01; *** *p* < 0.001.

**Table 3 animals-13-00983-t003:** Pearson correlation coefficients between physicochemical parameters of ewe’s Colostrum.

	PUFA	MUFA	SFA	∑ ω-3	∑ ω-6	L*	a*	b*
PUFA	1							
MUFA	−0.21	1						
SFA	−0.66 ***	−0.60 ***	1					
∑ ω-3	0.91 ***	−0.16	−0.63 ***	1				
∑ ω-6	−0.58 ***	0.47 ***	0.11	−0.60 ***	1			
L*	−0.15	−0.12	0.21	0.01	−0.15	1		
a*	0.20	−0.28 *	0.05	0.15	−0.14	0.12	1	
b*	0.18	0.31 **	−0.39 **	0.15	−0.06	−0.41 ***	−0.45 ***	1

PUFA: Polyunsaturated fatty acids; MUFA: Monounsaturated fatty acids; SFA: Saturated fatty acids; L*: Lightness; a*: Redness; b*: Yellowness; * *p* < 0.05; ** *p* < 0.01; *** *p* < 0.001.

**Table 4 animals-13-00983-t004:** Stepwise regression with Brix as dependent variable.

Step	Variable Entered	Partial R^2^	Model R^2^	F Value	<*p*
1	Protein	0.8054	0.8054	240.03	0.0001
2	Na	0.0131	0.8185	4.11	0.0472
3	Mg	0.0188	0.8373	6.47	0.0137
4	Ca	0.0301	0.8674	12.49	0.0008
5	Lactose	0.0138	0.8812	6.29	0.0152
6	b*	0.0098	0.8911	4.79	0.0331
7	∑ ω-6	0.0107	0.9018	5.68	0.0209
8	K	0.0065	0.9083	3.62	0.0626

**Table 5 animals-13-00983-t005:** Rotated Factor pattern.

Trait	Factor 1		Factor 2		Factor 3		Communality
	“Colostrum Quality of Grazing Ewes”		“Good Quality Colostrum”		“Mineral Component of Colostrum”		
Brix	−0.067		0.888	x	0.140		0.812
PRT	−0.022		0.974	x	0.005		1.000
UREA	−0.159		0.772	x	0.233		0.676
FAT	0.157		0.405	x	−0.028		0.189
LACT	−0.126		−0.420	x	−0.024		0.193
SCC	0.492	x	0.233		−0.071		0.302
pH	0.564	x	−0.079		−0.134		0.342
Ca	0.048		0.179		0.983	x	1.000
P	−0.077		0.216	x	0.031		0.054
K	−0.192		0.033		0.653	x	0.464
Mg	−0.222		0.419	x	0.725	x	0.750
Na	−0.053		−0.057		0.333	x	0.117
PUFA	0.970	x	0.096		−0.043		0.952
MUFA	−0.238		0.332	x	0.073		0.172
∑ ω-3	0.939	x	−0.008		−0.040		0.883
∑ ω-6	−0.610	x	0.127		0.165		0.416
L*	−0.054		−0.497	x	0.159		0.275
a*	0.215		−0.320	x	−0.226		0.200
b*	0.122		0.800	x	0.022		0.655
Variance explained	68.4%		23.3%		8.3%		

PRT: protein; LACT: Lactose; SCC: Somatic Cell Count; PUFA: Polyunsaturated fatty acids; MUFA: Monounsaturated fatty acids; L*: Lightness; a*: Redness; b*: Yellowness; x high loading values.

## Data Availability

Not applicable.

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
