# Peer review of "Chemometric Approaches to Analyse the Composition of a Ewe’s Colostrum"

_animals, 2023, doi:10.3390/ani13060983_

Round 1

Reviewer 1 Report

Dear authors,

Very well written paper on a highly important issue. Your team clearly knows what they are talking about. This work is of great value to sheep rearing in general.

There are some (small) items to addres.

Abstract

Line  19               Incorrect use of the word “fetus”. A fetus is the unborn offspring that develops from an animal embryo.

Line  25               First use of SCC, please specify somatic cell count (SCC) for a potentially lesser informed reader.

Line  26               parameters for (sub)clinical mastitis. Sub between brackets.

Line  33               First use of PUFA, please specify. Just as FA. Same for MUFA in line 36.

Introduction

Line  50               Presumably “ours” should be “hours”? First hours after lambing?

Line  65               What is meant by “high technical times”?

The introduction is very well written. To the point and all important issues addressed. Also the  scientific “gap” (to address the indicators in sheep colostrum) is clearly stated and how the researchers are adressing these issues and positioning against earlier work is clear.

M&M

Line  88               Out of curiosity why was the colostrum filtered? (Cleanliness?) And how? Also were gloves used when hand milking?

Line 122             NaOCH3 should be NaOCH3

Check for consistency 50 °C or 5°C.

Results

Line 204             (to be continued)?

Line 207 en 2013            Dot at the end. And 213 enter.

Lines 218-223    Unclear. Sentence too long. Suggestion to break into two sentences. One on correlation with colour and one Brix.

Line 255             Dot at the end. And enter.

Lines 256-274    Are interpretations. But I see why positioned here although it could be augmented that some items belong to discussion. No further action required.

Discussion

Lines 298-301    Do the authors agree on this assumption? Can you think of other explanations? Your paper is very well established on chemical parameters, is there a possible suggestion?

Lines 310-318    In the FA and the CIE LAB discussion you compare to previous work and use your new data. And come up with possible explanations. This is a really good part of the discussion. (No further action required.)

Line 331             Do the authors have a suggestion to explain the difference between ewes , cows and goats milk in regards to the negative/positive correlation between protein and SFA?

Line 369             Again what is meant by “rubbing” is this between the legs? Or with vegetation? And how would this indicate possible (subclinical) mastitis?

Line 385             Delate two, or make three.

Line 413-416     “will alow”? Seems like two sentences melded together. Rephrase. Since this possible intervention is also mentioned in the conclusion, please elaborate.

Was there ethical improvement for this work? (The hand milking part? Since I don’t think this is general practice?)

Reviewer 2 Report

Comments on manuscript 2237331

The manuscript entitled ‘Chemometric approaches for analyze the composition of a ewe’s colostrum’ investigated different statistical approaches efficiently summaring the ewes colostrum profiles. The findings will be benefical to provide the correct nutrition for the lambs. The content of this manuscript is fit for Animals. The manuscript is well prepared.

Several concerns on the current version.

1. A total of 60 samples from two seasons were selected in this study, authors have also declared that season has a significant effect on constituents of colostrum, however, all 60 samples were not divided into two groups to do further analysis, season was not considered in all Multivariate analysis. This is not reasonable, and the accuracy of the results is affected.

2. The Title should be modified. Chemometric approaches in the title? While more contents are multivariate analysis and to find the new variables explanning colostrum profiles.

3. The results of Pearson correlation analysis are too many and too confused, please have a focused description and discussion.

4. Eight most representative variables were selected in the SDA analysis, please give the reasons.

5. Discussion should be further revised. Discussion 4.2 were about the relationship between colostrum parameters, but is it a bit off topic to spend half of contents to discuss the relationships among breast health parameters? 

6. Are the concentration of milk fat percentage (8.01%) and milk protein percentage (12.10%) in colostrum so high? Or else which was due to the breed of ewe? Is the sheep selected a special dairy sheep breed?

7. The Standard deviation of Phosphorus content was very high, whats the reason?

8. Whats the specific fatty acids including in SFA, MUFA and PUFA? Should be clarified.

9. Generally, * p<0.05 and ** p<0.01 were widely used in the statistical analyses, *** p<0.001 was not the standard significant threshold in the statistics, which was not used.

10. The authors should explain and discuss all the significant Pearson correlation results? Why the two or multiple indexes existing correlations?Please make an explaination for all biological problems.

11. Whether the determination of milk composition is still accurate after adding 1:1 water to colostrum?

12. Please add criteria for statistical analysis of significant results in the statistical analysis section.

13. Please provide the full name when the abbreviation appears for the first time in the manuscript.

14. Simple statistics’?Should be Descriptive statistics results.

15. ‘Brix refractive index’ or ‘refractive Brix index’?

16. The language needs further polishing, such as ‘The aim of this survey was to investigate the relationships between ewes’ colostrum parameters applying a multivariate approach’?The objective of this study was to evaluate the correlation patterns between the bulk tank colostrum parameters and to define, using a multivariate approach, the relationships between them’?‘These constituents are also much more prominent in the first ours from lambing’?‘NaOCH3’?

17. Yellow index?

18. Limited references from the last five years were cited in the manuscript, please replace some very old references and cite new ones. 

Reviewer 3 Report

In my opinion, the self-citation of point 22 is done incorrectly: Todaro, M.; Scatassa, ML; Gannuscio, R.; Vazzana, I.; Mancuso, I.; Maniaci, G.; Laudicina, A. Effect of lambing season on colostrum composition of 490 sheep. Italian Journal of Animal Science 2022, in print in lines 75, 139, 158 and 295. In the other lines where this publication was used, the citation is, in my opinion, done very well, so please correct to a similar form as in lines 69, 313, 367, 399.

Reviewer 4 Report

Comments and Suggestions for Authors

In this paper, the authors investigate the relationships between ewes’ colostrum parameters applying a multivariate approach. The objective of this study was to evaluate the correlation patterns between the bulk tank colostrum parameters and to define, using a multivariate approach, the relationships between them. The authors demonstrated that the different statistical approaches used in this study were able to efficiently summarize the ewes’ colostrum profiles with a reduced number of new variables.

In fact, they showed the positive correlation between the Brix value and protein (0.90). Both parameters (protein percentage and Brix value) were then positively correlated with the percentage of fat, urea, calcium, and magnesium, as well as the yellow index (0.78 and 0.75). Stepwise discriminant analysis showed that the protein percentage, calcium, and magnesium were able to explain more than 85% of the Brix refractive index. Their study confirmed that the Brix refractive index is a good parameter for simply evaluating the nutritional quality of sheep colostrum on a farm. Understanding the correlations between the compositional parameters will allow for interventions to correctly feed lambs in their first days of life. The experiment was balanced with a sufficient number of bulk colostrum samples. The methods of analysis are good, clear and well detailed. The manuscript was well written.

Tables 1, 2.a and 2.b: I suggest:

Σw-3 Instead of ΣΩ-3. Ω refers to ohm, the unit of resistance of a material to the flow of electric current.

Line 244: the only probability is 0.05 and not 0.15. The probability is noted p and not P, to be corrected in table 3.

 Line 425: I suggest:

“the colostrum samples allowed to” Instead of “the colostrum samples allowed us to”

Round 2

Reviewer 2 Report

1. 'The significance threshold for entry into the model was p=0.15.'? Do you have evidence for threshold?Why not 0.05 or 0.01?

2. 'Also the positive correlation between a* and SCC or pH...' ? the * in the sentence was no needed.

3. There are so many significant Pearson correlation results. The authors indicate they have explained  the most indicative ones related to the biological factors. While the possible reasons for the two or multiple indexes existing correlations should be added in the Discussion section. 

4. The authors prefer to leave 'Brix refractive index', please keep it the same in the whole manuscript.
